# The Effects of Heat Treatment on Microstructure and Mechanical Properties of Selective Laser Melting 6061 Aluminum Alloy

**DOI:** 10.3390/mi13071059

**Published:** 2022-06-30

**Authors:** Wei Liu, Shan Huang, Shuangsong Du, Ting Gao, Zhengbin Zhang, Xuehui Chen, Lei Huang

**Affiliations:** 1School of Mechanical and Electrical Engineering, Anhui Jianzhu University, Hefei 230601, China; weiliu8202@163.com (W.L.); hs18556919674@163.com (S.H.); ahjzu_gt@126.com (T.G.); zhangzb0314@126.com (Z.Z.); huang175@ahjzu.edu.cn (L.H.); 2Key Laboratory of Intelligent Manufacturing of Construction Machinery, Anhui Jianzhu University, Hefei 230601, China; 3Institute of Plasma Physics, Chinese Academy of Sciences, Hefei 230031, China; ssdu@ipp.ac.cn

**Keywords:** selective laser melting, 6061 aluminum alloy, heat treatment, microstructure, mechanical properties

## Abstract

Selective laser melting technology can be used for forming curved panels of 6061 aluminum alloy thermal shield devices for the International Thermonuclear Experimental Reactor (ITER), in order to make the formed parts with better performance. This study proposes different heat treatment processes, including annealed treatment at 300 °C for 2 h, solution treatment at 535 °C and then aging at 175 °C over 2 h, to control the mechanical behavior of the 6061 aluminum alloy samples prepared by selective laser melting (SLM). The mechanical properties such as ductility, tensile strength, and hardness of SLM 6061 aluminum alloy were investigated, and the microstructure of the samples was analyzed. The eutectic silicon skeleton shape disappeared after annealing treatment at 300 °C for 2 h. The tensile strength decreased by 22.86% (from 315 MPa to 243 MPa of the deposited state samples), and the elongation increased from 2.01% to 6.89%. Moreover, the hardness reduced from 120.07 HV_0.2_ to 89.6 HV_0.2_. After solution aging, the unique microstructure of SLM disappeared. Furthermore, the precipitation of massive Si particles on the α-Al matrix increased, and a trace amount of the Mg_2_Si(β) phase was generated. Compared with the deposited samples, the tensile strength decreased by 12.06%, while the hardness of specimens was 118.8 HV_0.2_. However, the elongation showed a remarkable increase of 297% (from 2.01% to 7.97%). Therefore, solution aging can critically improve the plasticity without losing significant tensile stress in the SLM 6061 aluminum alloy. This study proposes the use of SLM 6061 aluminum alloy for the thermal shields on the ITER and provides a reference for choosing a reasonable heat-treatment method for the optimal performance of the SLM 6061 aluminum alloy.

## 1. Introduction

Thermonuclear Experimental Reactor (ITER), as a large international scientific engineering project to study thermonuclear fusion, aims to provide the most ideal clean new energy for mankind. The overall structure of ITER device mainly includes superconducting magnet system, vacuum chamber, thermal shield, dewar and support structure. As one of the key components, the thermal shield is located between the superconducting magnet and dewar. Its main function is to reduce the heat load transferred to the superconducting magnet, so as to further ensure the low temperature environment of the superconducting magnet system and ensure the stability of the superconducting magnet during operation. Due to the large volume of cold screen, the mechanical properties of cold screen are required. At the same time, in order to ensure the cooling effect of the cold screen, the thermal conductivity of the cold screen material should be high. Aluminum alloys have been utilized in thermal shield devices for the International Thermonuclear Experimental Reactor (ITER) due to their low density, suitable corrosion resistance, high specific strength, and excellent thermal conductivity [1,2,3,4,5]. Although traditional manufacturing methods have been used to fabricate thermal shields, there are still some disadvantages in the production and application. In general, the major problem is the complexity of thermal shield components, such as curved panels and ductwork. These components are difficult to manufacture and assemble, and most of them need to be machined on CNC machines to meet tolerances. In addition, the mechanical properties of the thermal shields formed by the traditional aluminum alloy manufacturing method are not good enough. Consequently, a thermal shield forming method with high dimensional accuracy and near-net-shape is a development target for the future [6,7].

Selective laser melting (SLM) is considered one of the most rapidly developing manufacturing methods. The schematic of layer-by-layer laser melting during the SLM process is shown in Figure 1. Based on the principle of discrete accumulation, similar to other AM technologies, SLM technology is expected to break the cycle of design and production and become a method with high flexibility and short time circle, thus completely changing the traditional manufacturing methods. SLM technology uses a high-energy laser beam to melt fuse powders according to predetermined tracks, and components are manufactured layer by layer [8]. When a layer is irradiated by the laser beam, the layer is melted and solidified at a high rate. After the layer is built completely, some defects inevitably appear in the present layer. These defects, like thermal cracking and residual stress concentration, usually appear during the SLM forming process of aluminum alloys, which affects the mechanical properties of the formed specimen. These drawbacks limit the application of the SLM technology into aluminum alloys [9].

As one of the critical methods of enhancing the microstructure and mechanical properties of SLM aluminum alloy, heat treatment has gradually attracted the attention of many scholars. Siyu Sun et al. [10] studied the microstructure, hardness, and phase transformation of SLM-prepared 7075 aluminum alloys under the solid solution + double aging heat treatment method. The study showed that the specimens had better microstructure and properties: The solid solution temperature reached 470 °C, and the double-aging temperatures were 110 °C and 150 °C. The authors of [11] investigated the effect of solid solution aging treatment on the mechanical properties of SLM-formed AlSi10Mg alloy specimens and found that controlling the solid solution temperature can improve the tensile properties and microhardness of SLM-formed AlSi10Mg alloy specimens. Tianchun Zou et al. [12] studied the effects of different process parameters and heat-treatment parameters on the microstructure and mechanical properties of AlSi7Mg prepared by SLM. The results showed that with increasing annealing temperature, the tensile strengths all decreased, while elongation increased. With increasing solution temperature, the tensile strength and elongation increased at first and then decreased. The fracture mode of the specimen is brittle and ductile-mixed fracture. The fracture morphology of the specimens after heat treatment is large and deeply dimpled, showing the toughness pattern. L.F. Wang et al. [13] investigated the enhancement of mechanical properties of SLM-formed AlSi10Mg alloy specimens by T6 heat treatment. The observation indicates that T6 heat treatment can eliminate anisotropy and enhance the ductility of the samples without significant loss of tensile strength. A.B. Spierings et al. [14] studied the effects of different heat-treatment temperatures and holding times on the mechanical properties and microstructure of Al-Mg alloy modified with Sc and Zr, and the results showed that the static mechanical properties are exceptionally good, with Rm values exceeding 500 MPa, along with almost no build-orientation related anisotropic effects and a high ductility even in the heat-treated condition. Scudino et al. [15] studied the wear behavior of Al-7Si-0.5Mg-0.5Cu alloy prepared by SLM under heat treatment. The results show that the specific wear rate of SLM Al-Cu-Mg-Si alloy after T6 heat treatment is lower, but the average friction coefficient is similar to that of Al2024 alloy after heat treatment. P. Ponnusamy et al. [16] investigate the dynamic behavior of SLM-processed and heat-treated eutectic AlSi12 alloy under compression. The results showed that a significant reduction in dynamic yield strength and ultimate compressive strength was observed when the as-built AlSi12 samples were tested at elevated temperatures (200 °C and 400 °C).

In summary, the aluminum alloy materials formed by SLM have high sensitivity to heat treatment, and the comprehensive mechanical properties of SLM-formed aluminum alloys can be enhanced through reasonable heat-treatment processes. At present, the research on the microstructure and mechanical properties of aluminum alloys prepared by SLM mainly focuses on AlSi10Mg [13], Al12Si [17], Al-Cu [18], and Al-Zn systems [10]. Among the SLM-forming aluminum alloy series, SLM 6061 aluminum alloy is used in part of the thermal shield device in the ITER due to its high strength at low temperature, good processing performance, and strong corrosion resistance. In addition, due to the large volume of cold screen parts, the mechanical properties of the materials are required to be high after the cold screen parts are formed. Based on these varied findings, it is significant to study the properties of 6061 aluminum alloy for SLM fabricated parts.

Therefore, in order to enhance the properties of SLM 6061 aluminum alloy, it is meaningful to investigate the effects of different heat-treatment methods on its microstructure and mechanical properties. This paper investigates and discusses the influence of annealed and solution-aging heat-treatment methods on the microstructure and mechanical properties of SLM 6061 aluminum alloy. The samples formed by SLM were deposited, annealed at 300 °C for 2 h, and then treated in a solid solution at 535 °C for 1 h followed by an artificial aging treatment for 2 h at 175 °C. Then, the effect of heat treatment on the microstructure and mechanical properties of the specimens was analyzed. 

## 2. Experimental Materials and Methods

### 2.1. Experimental Materials and Sample Preparation

The 6061 aluminum alloy powders provided by SimpNeed. Powders were dried in a drying oven at the temperature of 100 °C in order to remove the residual moisture before the selected laser melting manufacturing. The chemical composition of SLM 6061 aluminum alloy powder is shown in Table 1. The surface morphology of the SLM 6061 aluminum alloy powder was observed by scanning electron microscopy, as shown in Figure 2a. The powder particles are mainly spherical and near spherical, with excellent sphericity and fluidity, which is conducive to the flatness and spreadability of the powder layer. The laser particle analyzer measured the particle size of the SLM 6061 aluminum alloy powder. It can be seen from Figure 2b that the powder particle size distribution is normal, mainly in the range of 10 to 30 μm. In Figure 2b, the typical laser particle sizes of powder D10, D50, and D90 are 12.995, 20.289, and 33.408 μm, respectively. Moreover, the average particle size is 20.2889 μm. Because the thickness of the powder layer was 30 μm, the average particle size of the powder is of the same order of magnitude as the thickness of the powder layer. It is smaller than the thickness of the spreading layer, which guarantees the effect of spreading powder.

The SLM equipment used in the test was the M350 version by SimpNeed. The laser power (*P*) was 503 W, the scanning speed (*v*) was 1000 mm/s, the layer thickness (*t*) was 30 μm, and the hatch spacing (*h*) was 150 μm. As shown in Figure 3a, the scanning strategy is a rotation angle (*θ*) of 345° for the adjacent layer along the forming direction. In line with the experimental necessities, 3 groups of 18 samples were prepared. They are the deposited group, the annealing group, and the solution aging group. The forming direction and dimension parameters of tensile sample are shown in Figure 3b. The density of samples with different heat treatment conditions was measured by the Archimedes drainage method. The density test equipment is a ZMD series electronic density instrument model BL-05-SY-143. In order to ensure the accuracy of the density testing process, six samples were selected from each group, and the average density of three samples was taken as the final density for the group. The theoretical bulk density of the 6061 alloys is 2.75 g/cm^3^. After SLM, the densifification of the as-fabricated samples was 95.3%. The specific value of density is 2.622 g/cm^3^, that of the annealed sample was 2.632 g/cm^3^, and that of the solution-aged sample was 2.626 g/cm^3^.

### 2.2. Test Method

#### 2.2.1. Heat Treatment and Microstructure Observation

The heat treatment of SLM specimens was performed in a JS-SHPB-20-NA-C/H(H) vacuum furnace. Before the experiment, the resistance furnace was evacuated to isolate the air. The specific two heat-treatment processes are summarized in Table 2. The stress relief annealing procedure was applied: The specimen was heated to 300 °C at a rate of 10 °C/min and held for 2 h, and thereafter changed into cooled to temperature of 25 °C with the furnace. The solution treatment was carried out at the rate of 10 °C/min to 535 °C, held for 1 h, then artificially aged at 175 °C for 2 h. Finally, the specimens were cooled to 25 °C by water cooling. The schematic schedule of the heat treatment is shown in Figure 4.

The specimens were divided into three groups according to the state: deposited, annealed, solid solution. The cross and longitudinal sections of the heat-treated specimens were polished by 400#, 800#, 1200#, and 1500# sandpaper to obtain a bright mirror surface with suitable reflectivity. Subsequently, the polished samples were etched by Keller’s reagent for 20–60 s to observe the different phase microstructures. The phase composition of the specimens was analyzed by X-ray diffractometer with diffraction angles ranging from 30° to 90° in steps of 10 °/min. The melt pool morphology of the polished samples was observed by Axiovert 40 MAT optical microscope. The microstructure of specimens was analyzed using a Zeiss EVO180 scanning electron microscope. The elemental distribution was analyzed using a Zeiss Gemini SEM 500 scanning electron microscope.

#### 2.2.2. Mechanical Performance Test

Mechanical tests of the as-fabricated and heat-treated samples involve the evaluation of tensile strengths, plasticity, and hardness. The uniaxial tensile test was conducted on WDW-300A electronic universal testing machine. Before the test, the reference length of the labeled specimens is 23 mm, and the tensile parameters of the specimen are set: the crosshead speed is 0.2 mm/min. The stress–strain diagram of the specimen can be obtained after being pulled, according to which the tensile strength can be calculated, and then the length after being pulled can be measured. At this time, the fracture elongation of the specimen can be calculated according to Equation (1):(1)$$\delta_{1}=\frac{L-L_{0}}{L_{0}}\times 100\%$$

Here, *δ_1_* is the fracture elongation of the specimen; *L* is the length L after being pulled; and *L_0_* is the the length of the calibration.

The tensile route was parallel to the forming direction of the SLM specimen. The tensile specimen dimensions are shown in Figure 3b and placed in SEM to observe the fracture morphology. All tensile examinations were carried out at room temperature. The hardness of the alloy was measured on the mirror surface of specimen with a Wilson Hardness 401MVD microhardness tester, which is set at a load of 200 g with holding time of 15 s. The Vickers hardness was calculated using the mean of five measurements conducted on a single sample.

## 3. Experimental Results and Analysis

### 3.1. Physical Phase Analysis

The XRD patterns of the SLM 6061 aluminum alloys in different states are depicted in Figure 5. It can be seen that the diffraction peaks of Al and Si phases are mainly presented in the deposited-state samples. The intensity of the Al phase diffraction peak is significantly higher than that of Si, indicating that the content of the Si phase in the deposited state specimens is shallow. After annealing treatment, the alloy’s diffraction peaks of Al and Si phases are still present. The intensity of the diffraction peak of the Si phase is enhanced. In addition, a small amount of AlFeSi phase diffraction peaks were observed, but no Mg_2_Si phase diffraction peaks were present. AlFeSi phase also appeared in the solution-aging state samples. Moreover, a faint Mg_2_Si (β) phase was observed in the solution-aging state. During the preparation of SLM 6061 aluminum alloy, Si and Mg are solidly dissolved in the Al matrix under the effect of a high cooling rate to form a supersaturated solid solution. As a result, only two main phases of Al and Si were present in the deposited-state alloy. The solution-aging treatment promotes the precipitation of the elements. However, after the solution-aging treatment, the Mg content of the alloy is only 0.3%, and the amount of precipitated Mg_2_Si phase is trace. The remaining Si atoms continue to precipitate to shape bulk Si particles. Therefore, the Si phase diffraction peak intensity after the solution aging was higher than that of the deposited and annealed states.

### 3.2. Microstructure Analysis

Figure 6 shows the metallographic microstructures of the deposited, annealed, and solution-aged state alloys. Among them, Figure 6a–c are OM images of the X–Z surfaces of the different state samples. Figure 6d–f are OM images of the X–Y surface of the different states SLM-prepared samples. The results show no significant difference between the metallographic morphology of the deposited and annealed alloys. The morphology of the melt pool intertwined after solidification can be observed in the X–Y plane. The analysis concluded that the interwoven morphology of the molten pools was caused by the intersection of the scanning tracks layer by layer. The depth of the center of the melt pool was greater than the edge of the melt pool [19]. In the same specimen, the melt pool distribution is not uniform due to the SLM-forming method of rotating the adjacent layers by 345°, resulting in overlapping scanning tracks. Moreover, there were 5–10 μm pores in the adjacent melt track of the deposited state specimen (the black arrow in Figure 6a indicates the position of the pores). Such defects are due to the forming process of residual gas in powders escaping from the melt pool incompletely. In addition, the laser energy density (LED) has a certain inhibition effect on the pore defects. The LED represents the energy of the laser acting on the powder per unit time. The higher the LED, the better the melting performance of the powder, but the excessive increase in LED can have detrimental effects as well. Laser power, scanning speed, scanning distance, and powder layer thickness all determine the LED. The calculation formula of LED is as follows [20]:(2)$$X_{LED}=\frac{P}{vht}$$
where *X_LED_* (J/mm^3^) is the LED, *P* (W) is the laser power, 503 W; *v* (mm/s) is the scanning speed, 1000 mm; *h* (mm) is the hatch spacing, 150 mm; and *t* (mm) is the layer thickness, 0.03 mm.

After calculation, the LED is 111.78 J/mm^3^. It is much larger than the 63 J/mm^3^ in [21]. Combined with the density test results above, it can be seen that the density of the deposited sample decreases significantly, resulting in an increase in the number of pores.

It can be seen from Figure 6c,f that the blurry melt track on the X–Y surface of the SLM preparation specimen after the solution treatment at 535 °C for 1 h and aging treatment at 175 °C for 2 h, and the fish-scale shape of the melt pool on the X–Z surface disappears. In addition, the unique microstructure of SLM is replaced by a large number of massive Si particles precipitated on the α–Al matrix.

Figure 7, Figure 8 and Figure 9 show the microstructure of the deposited, annealed, and solution-aging state alloys observed under scanning electron microscopy near the melt pool boundary and the EDS energy spectrum analysis results of some points. It was observed that there is a prominent skeleton structure in the deposited state alloy. According to the EDS analysis of the deposited state (Figure 7), the Si content in the gray area is higher than that in the black area wrapped in the middle. From this, it is inferred that the gray area is the Si framework of near-eutectic composition, and the black area wrapped in the middle is the α–Al matrix. The reason for eutectic silicon skeleton formation is related to the cooperative growth mode of the solidification process of Al–Si eutectic. Si crystals constantly arrange and alternate the growth direction as they solidify in this mode. During the solidification process, α–Al is preferentially nucleated, and a large amount of Si element is dissolved in α–Al to form near-supersaturated solid solution under the condition of high cooling rate of SLM. Due to the low melting point of Si, supersaturated Si diffuses outside of α–Al. In the process of growing up, neighboring Si encounter each other and become embedded in α–Al solid solution, forming an interconnected skeleton structure in space [21].

According to Figure 8, the eutectic Si skeleton structure disappeared after the annealing treatment. Compared with the EDS results for the deposited state (Figure 7), the Si composition in the white region increased after annealing, which indicated that the annealing heat treatment accelerated the growth rate of Si crystals. Long needle-like precipitates can be observed on the surface of the alloy after solution-aging treatment (Figure 8). After the EDS analysis, the long needle-like depositions contain Fe elements, and the XRD test results indicate that the precipitates are AlFeSi-β phase [22]. It can be observed that the bulk depositions contained 77.97% Si. As shown in the dashed line (Figure 9), the solution-aged specimen contains more pores compared with other states. The formation is mainly due to the pores gathered during the high-temperature heat treatment.

### 3.3. Tensile Performance Analysis

The mechanical properties of specimens before and after heat treatments were shown in Figure 10. For each test condition, three samples were tested and a representative curve that is selected from two best consistent stress–strain curves is presented. It can be observed that the tensile strength and elongation of the traditional hot-rolled 6061 aluminum alloy sample are 182 MPa and 13.40%, respectively. The tensile strength and elongation of the deposited state samples prepared by SLM were 315 MPa and 2.01%, respectively. Compared with the hot-rolled sample, the tensile strength of SLM fabricated parts increased by 73.08%, but the elongation reduced from 13.04% to 2.01%. After different heat-treatment methods, the plasticity of the SLM-prepared samples was significantly improved, but the strength was quite different. After annealing heat treatment at 300 °C for 2 h, the strength of the sample is 243 MPa, which is 22.86% lower than that of the deposited state. However, its elongation reached 6.89%. After solution aging, the strength of the sample is 277 MPa. Compared with the deposited samples, the tensile strength decreased by 12.06%, but the elongation showed a remarkable increase to 297% (from 2.01% to 7.97%). The mechanical properties of the samples are shown in Table 3. 

Figure 11 shows the SEM images of the fractures in the Z-directional tensile specimens of the alloys in the deposited state, the annealed state, and the solution-aging state. It can be seen from Figure 11a–c that there are obvious defects in the surface of the fracture in all three states. The different causes of defects can be divided into pores, micro-cracks, unmelted powder, and melting track. The difference in physical properties between the melt solidification region and the unmelted powder particles during solidification results in a weak bond between the melt solidification zone and the interface of the unmelted powder particles, and microcracks occur at the weak interface with the increase of tensile stress [21].

In the fracture morphology, the shape of pores caused by unmelted powder is generally round. After heat treatment, the fracture surface of the alloy is relatively smooth. The size of defects like pores and cracks increased, and the number of defects decreased. Most cracks were generated at the defect sites in the α–Al matrix throughout the fracture process. According to Figure 12a, a large number of holes, spherical powder and fusion channels are distributed on the fracture surface of the deposited alloy. From Figure 12a, it can be seen that the tensile fracture of the deposited specimen does not exhibit a clear dimple shape. This is mainly due to the fact that the eutectic Si skeleton divides the α–Al matrix, resulting in the crack expansion being hindered by the eutectic Si skeleton, which prevents the formation of a complete dimple shape in the α–Al matrix, hence the low plasticity of the specimens. During the fracture of the deposited specimens, most of the cracks arise in the defective areas in the α–Al matrix. However, the presence of eutectic Si skeleton morphology in the deposited state hinders the crack expansion. As the tensile stress increases, the cracks can only extend along the α–Al matrix or eutectic Si skeleton interface. This is one of the main reasons for the high strength of the deposited alloy. The fractures in the annealed and solution-aging specimens have only a few pores and microcracks and are generally flat. This is due to the cracking of the eutectic Si skeleton after heat treatment, and the crack source does not need to expand along the grain boundary. Therefore, the defect is given enough space to grow, so the fracture surface is relatively flat [23]. From Figure 12b, it can be seen that there are small dimples in the fracture of the annealed specimen, which shows the characteristic of ductile fracture. This also proves that the plasticity of SLM 6061 aluminum alloy increases after annealing, and the change in the shape of eutectic Si skeleton during the annealing process is one of the main reasons for the increase in plasticity. After annealing, the eutectic Si skeleton gradually fractures and coarsens. The loss of crack hindrance by the eutectic Si skeleton makes it easier for cracks to expand along the melt pool boundary between the stacked layers. This is the main reason the strength of SLM 6061 aluminum alloy specimens decreases after annealing. After solution aging, it can be seen from Figure 6 that the structural morphology of SLM disappears, so there is no residual crack to hinder the mechanical response, and the grain boundary dislocation increases. Therefore, the strength increases, and the ability to resist failure is enhanced [24]. It can be seen from Figure 12c that the size of the fracture dimple is larger than that of the annealed state, so its plasticity is the strongest.

### 3.4. Vickers Hardness Analysis

Heat treatment has a great influence on the microhardness of the sample [25]. It can be seen from Figure 6a,b that the surface distribution of SLM in deposited state and annealed state is uneven, and the molten pool presents fish scale distribution. Therefore, the hardness measurement results as deposited and annealed may be affected by the measurement area. The hardness of the weld pool boundary and the weld pool interior in the deposited and annealed states were measured respectively. Figure 13 and Figure 14 show the micro morphology of the inner and boundary hardness measurement of the deposited and annealed molten pool respectively. In order to ensure the accuracy of the test process, the hardness values of three different positions in the molten pool were measured. When the measuring area is the boundary of deposited molten pool, the average hardness value is 124.4 HV_0.2_. When the measuring area is inside the deposited molten pool, the average hardness value is 116.8 HV_0.2_ The average hardness of the as deposited sample is 120.6 HV_0.2_. The average hardness of the melten pool boundary is 93.5 HV_0.2_. The average hardness inside the molten pool is 85.7 HV_0.2_ The average hardness of the annealed sample is 89.6 HV_0.2_. Therefore, the hardness of the melten pool boundary is greater than that of the melten pool interior in both the deposited and annealed states. The measurement results of deposited and annealed state are shown in Table 4 and Table 5. It can be seen from Figure 6c that the molten pool morphology of the sample treated by solution aging disappears and the surface is relatively uniform. Therefore, six different positions of the sample are selected for hardness measurement, and the hardness of the sample in solution aging state is 118.8 HV_0.2_. Figure 15 shows the Vickers hardness values of the hot-rolled parts, SLM-deposited state, SLM-annealed state, and SLM solid-solution alloy specimens. The findings show that there is a strong relationship between heat-treatment temperature and the microhardness of the SLM specimens. It can be observed that the lowest hardness was at 77.57 HV_0.2_ in the hot-rolled parts. The hardness of the SLM-prepared samples was 120.07 HV_0.2_, which was greater than the hot-rolled prepared parts. The Vickers hardness decreased to 89.6 HV_0.2_ after annealing heat treatment at 300 °C for 2 h, 25.38% lower than the deposited-state alloy. In addition, the Vickers hardness of the specimens increased greatly compared with that of annealed state after the solution-aging heat treatment. The hardness of specimens after the solution-aging heat treatment reached 118.8 HV_0.2_, which was close to the deposited-state samples. However, the hardness of tensile specimens in the three stages is greater than the hardness of hot-rolled samples out of 6061 aluminum.

## 4. Conclusions

In this paper, the SLM 6061 aluminum alloy is proposed for the thermal shield device of the ITER project. To enhance the mechanical properties of the thermal shield panel, different heat treatments were applied to the SLM 6061 aluminum alloy specimens. We compared the microstructure, tensile strength, and Vickers hardness of the deposited, annealed, and solution-aging states of the SLM 6061aluminium alloy specimens. Based on the results of the study, conclusions can be summarized as follows:(1)The deposited microstructure contained Al matrix and eutectic Si framework. After annealing, a small amount of the AlFeSi phase appeared in the microstructure of the specimens. After solid solution, the α–Al phase, the Si phase, the AlFeSi phase, and the weak Mg_2_Si (β) phase can be observed in specimens. Moreover, the Si phase strength is higher than that of the deposited and annealed states because the remaining Si atoms continue to precipitate to form massive Si particles. In addition, the deposited samples and annealed samples have similar melt pool morphology. However, the unique melt pool morphology of SLM disappeared after solution aging.(2)After annealing, the typical microstructure features of the sample surface are Mg2Si phase precipitation and eutectic silicon skeleton fracture. As a result, the strength of the sample decreased significantly, only 243 MPa, although the elongation increased to 6.89%. The tensile strength of the solution-aged sample is 277 MPa. The hardness is 118.8 HV_0.2_ after solid-solution aging, which is close to the deposited state. In addition, the elongation reached 7.89%, which is 293% higher than the deposited state. Therefore, solution aging can significantly enhance the plasticity of the SLM-prepared 6061 aluminum alloy without sharply decreasing the tensile strength and hardness of the alloy.(3)Compared with the fractures in the deposited specimens and the annealed heat-treatment specimens, the fractures in the specimens after solution aging were flatter. The defects like pores and cracks were larger after solution aging. The dimples in the tensile fracture of the deposited specimens could not be clearly observed. After annealing, tiny dimples can be observed in the fractures in the specimens. In addition, the dimples in the fracture of the alloy in the solution-aging state are larger than those in the deposited and annealed states, which indicated that the solid-solution specimens have the highest plasticity of the three states.

## Figures and Tables

**Figure 1 micromachines-13-01059-f001:**
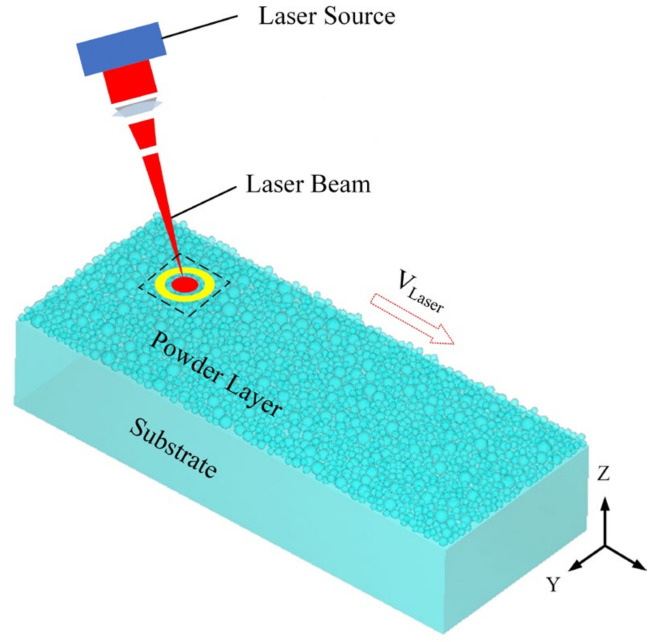
Schematic diagram of the SLM-forming process.

**Figure 2 micromachines-13-01059-f002:**
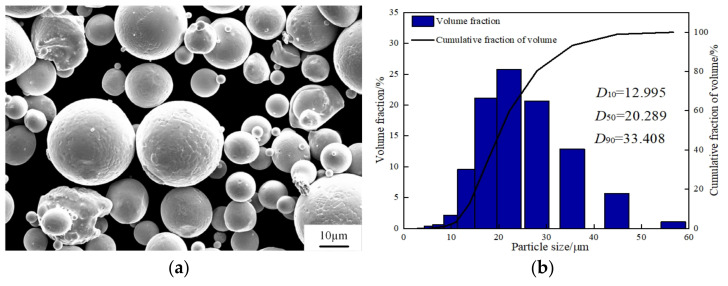
SLM 6061 aluminum alloy powders: (**a**) morphology; (**b**) size distribution.

**Figure 3 micromachines-13-01059-f003:**
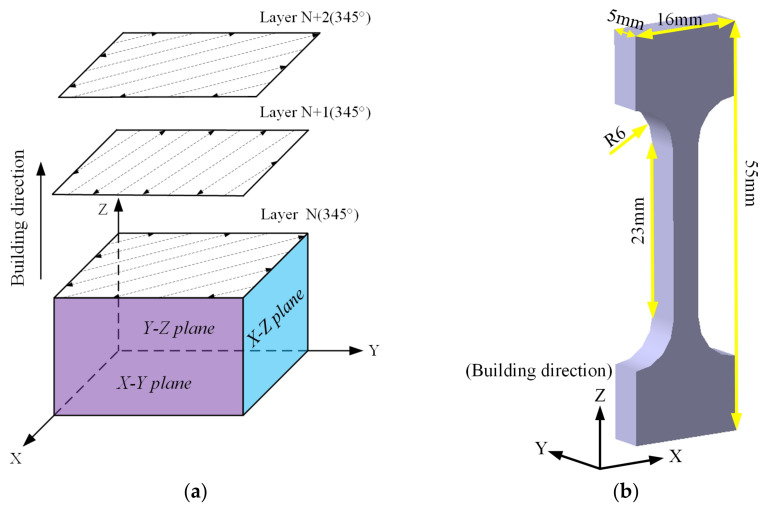
(**a**) Forming strategy; (**b**) building direction of SLM 6061 aluminum alloy.

**Figure 4 micromachines-13-01059-f004:**
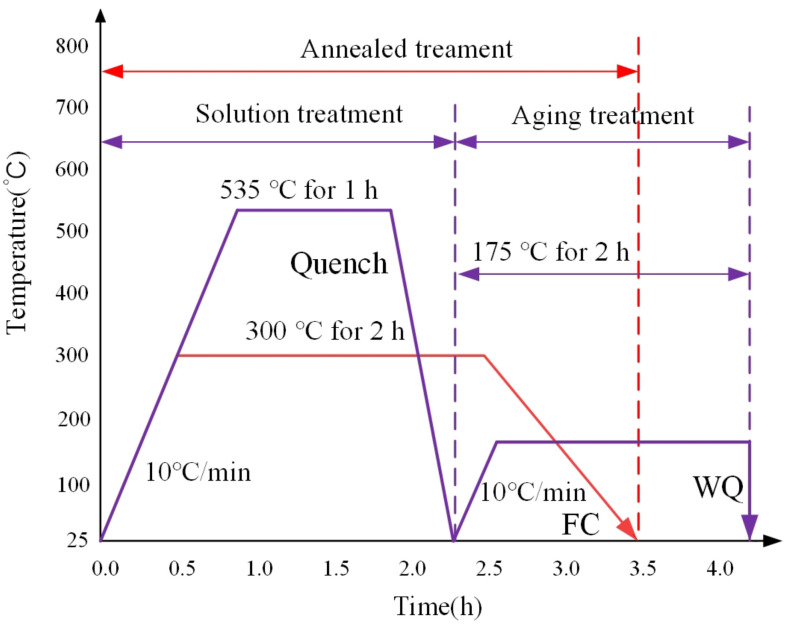
Schematic diagram of the heat-treatment process.

**Figure 5 micromachines-13-01059-f005:**
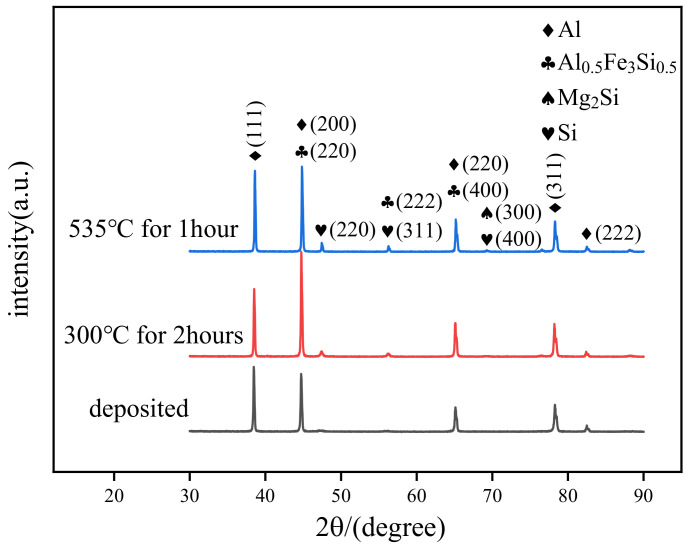
XRD curve and phase analysis.

**Figure 6 micromachines-13-01059-f006:**
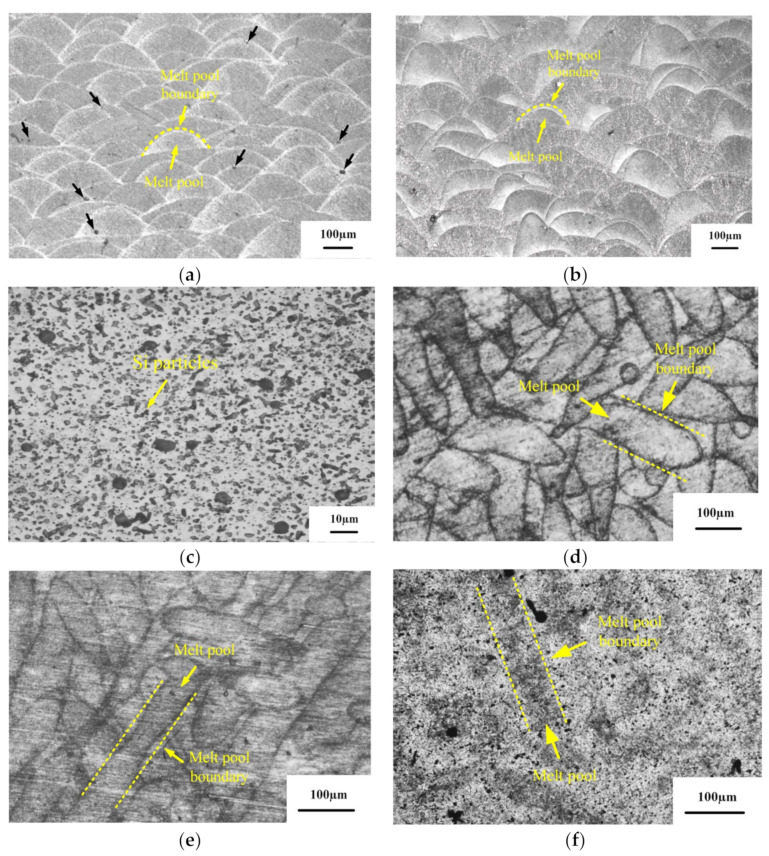
Microstructure of SLM 6061 aluminum alloy samples under different heat-treatment methods: (**a**) deposited, in X–Z direction; (**b**) annealing state, in X–Z direction; (**c**) solution and aging state, in X–Z direction; (**d**) deposited, in X–Y direction; (**e**) annealing state, in X–Y direction; (**f**) solution and aging state, in X–Y direction.

**Figure 7 micromachines-13-01059-f007:**
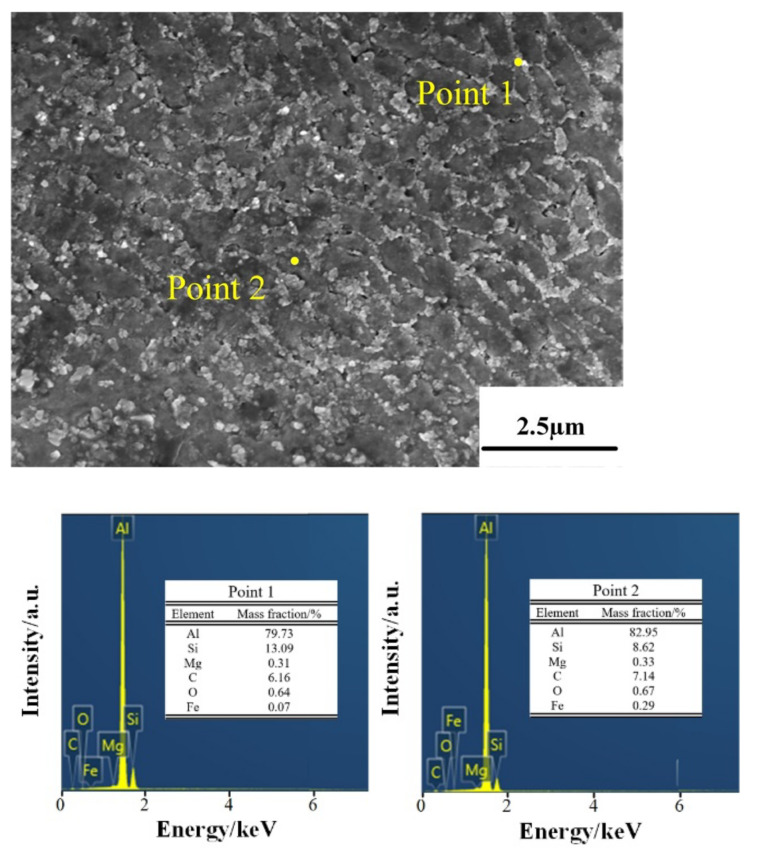
EDS results for the deposited samples.

**Figure 8 micromachines-13-01059-f008:**
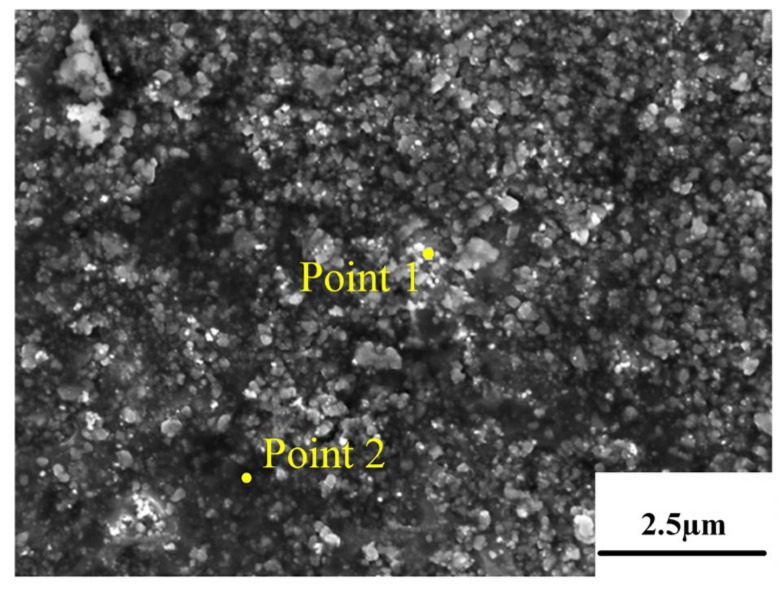
EDS results for the annealed sample.

**Figure 9 micromachines-13-01059-f009:**
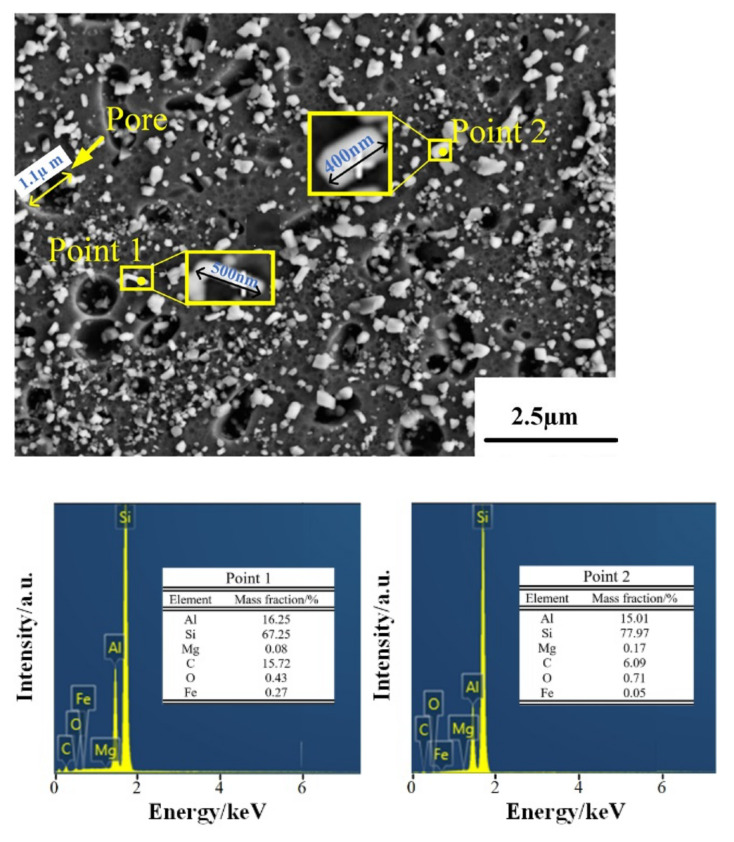
EDS results of solution aging sample.

**Figure 10 micromachines-13-01059-f010:**
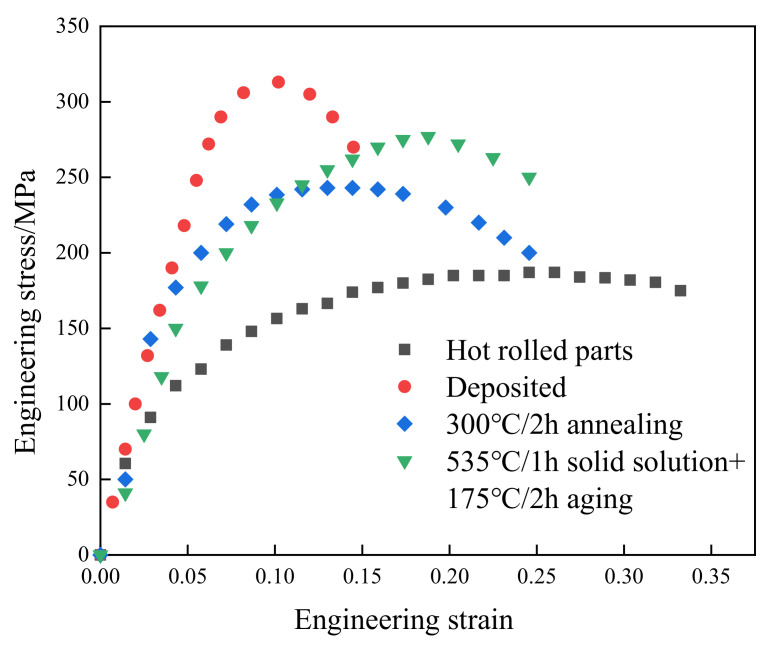
Engineering stress–strain curves for the samples.

**Figure 11 micromachines-13-01059-f011:**
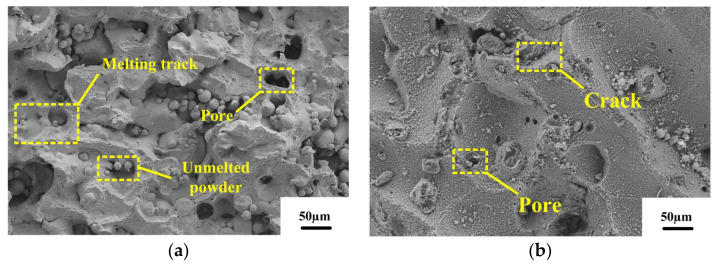
Tensile specimen fracture microscopic morphology: (**a**) deposited; (**b**) 300 °C/1 h; (**c**) 535 °C/1 h + 175 °C/2 h.

**Figure 12 micromachines-13-01059-f012:**
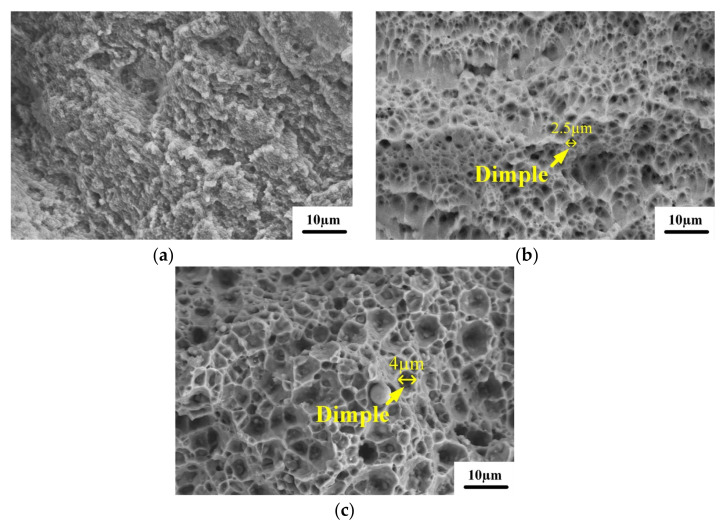
Microscopic morphology of tensile fracture under high magnification: (**a**) deposite; (**b**) 300 °C /1 h; (**c**) 535 °C /1 h + 175 °C /2 h.

**Figure 13 micromachines-13-01059-f013:**
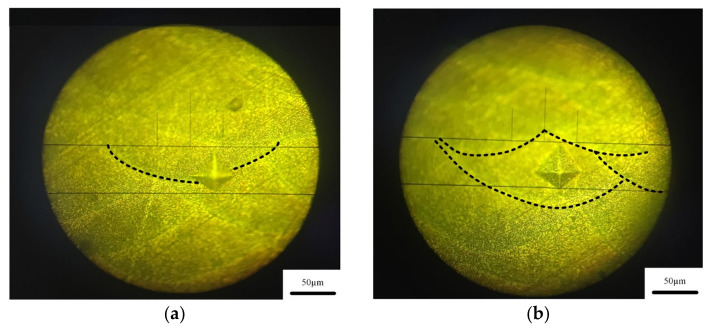
Microstructure of deposited state hardness test. (**a**) melt pool boundary, (**b**) melt pool.

**Figure 14 micromachines-13-01059-f014:**
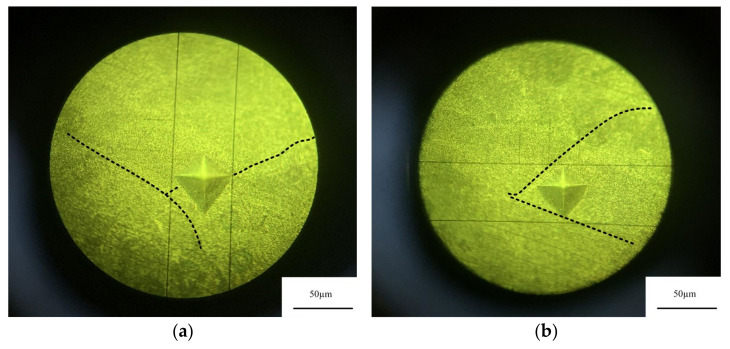
Microstructure of annealed state hardness test. (**a**) melt pool boundary, (**b**) melt pool.

**Figure 15 micromachines-13-01059-f015:**
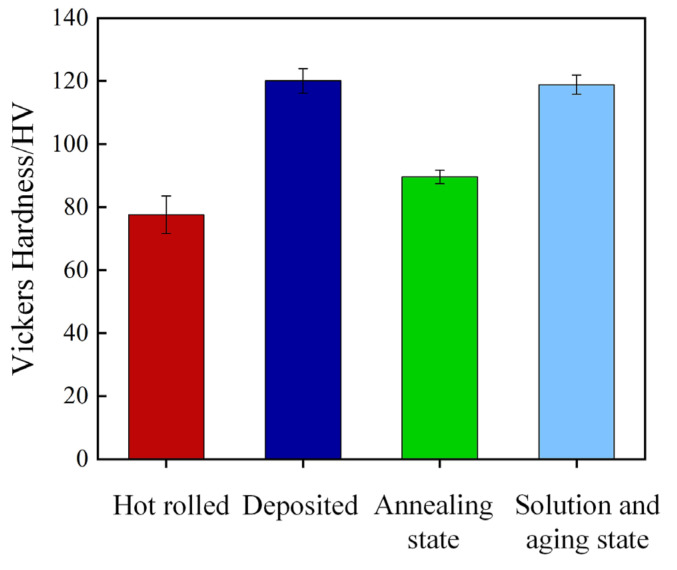
Average Vichers hardness of samples.

**Table 1 micromachines-13-01059-t001:** The chemical components of 6061aluminum alloy powder.

Composition	Al	Si	Mg	Cu	Mn	Zn	Fe	Ti	Cr
Mass fraction/%	bal	0.95	1.3	0.3	0.05	0.05	0.55	0.15	0.175

**Table 2 micromachines-13-01059-t002:** The processes of heat treating SLM 6061 aluminum alloy.

Number	Heat Treatment Process	Cooling Method
I	Deposited	-
II	Stress relief annealing at 300 °C for 2 h	Furnace cooling
III	Solution treatment at 535 °C for 1 h, then artificial aging at 175 °C for 2 h	Water cooling

**Table 3 micromachines-13-01059-t003:** Mechanical properties of the samples.

Sample	Ultimate Tensile Strength (Mpa)	Elongation (%)
Hot rolled	182	13.4
Deposited	315	2.01
Annealing state	243	6.89
Solution and ageing state	277	7.97

**Table 4 micromachines-13-01059-t004:** Hardness test results of deposited state.

Group	Hardness of Melt Pool Boundary/HV_0.2_	Hardness of Melt Pool/HV_0.2_
I	120.9	113.9
II	127.2	117.4
III	125.2	119.1
Average value	124.4	116.8

**Table 5 micromachines-13-01059-t005:** Hardness test results of annealed state.

Group	Hardness of Melt Pool Boundary/HV_0.2_	Hardness of Melt Pool/HV_0.2_
I	91.3	83.9
II	95.3	86.9
III	93.9	86.3
Average value	93.5	85.7

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
