# Peer review of "The Effects of Heat Treatment on Microstructure and Mechanical Properties of Selective Laser Melting 6061 Aluminum Alloy"

_micromachines, 2022, doi:10.3390/mi13071059_

Round 1
Reviewer 1 Report
Effects of Heat Treatment on Microstructure and Mechanical Properties of Selective Laser Melting 6061 aluminium alloys
Reviewer comments – June 2022
I am including the article as an annotated PDF, which contains many comments, suggestions and questions to the authors. I am including only general comments and suggestions here.
1 Overview of the topic
1.1 Relevance
The work is relevant, because the SLM manufacturing of aluminum alloys presents many problems and the selection of the adequate heat treatment for SLM – fabricated Al alloys is critical in attaining the best mechanical properties.
2 Analysis of the paper
2.1 Abstract
2.1.1 The abstract is mostly clear, as it presents a summary of motivations, methodologies and main conclusions. It mirrors the contents of the work without significant issues.
2.2 Introduction
2.2.1 The introduction starts by addressing the context of the problem, namely the thermal shield application. However, the relevance of the application itself is arguable, as it does not seem to influence the direction and results of the study. No relation is drawn between the component requirements and the properties sought for the material. Enhanced mechanical properties are relevant in almost any application, one could argue. Moreover, there is no depiction of the component, which leaves the reader somewhat confused.
2.2.2 The authors then describe the SLM process and associated issues, namely the defects. This is very relevant, especially in the case of aluminum alloys. This is demonstrated along the text by several references. But there seems to be a mixture between SLM manufacturing and Laser Re-melting during SLM manufacturing, which are not the same. So I would strongly suggest the authors to clarify this issue, especially since the Re-melting strategy does not seem to be used in this work.
2.2.3 Suggestions / corrections:
2.2.3.1 please review several of the references in the text, which seem out of place, or switched some how. See the comments on the PDF.
2.2.3.2 Please review the formatting of the figure 1 caption
2.3 Experimental materials and methods
2.3.1 Experimental materials and sample preparation: no major issues, please see the attached PDF for small corrections.
2.3.2 Test Method - Heat treatment and micro-structure observation: no major issues, please see the attached PDF for small corrections.
2.3.3 Test Method - Mechanical performance test: no major issues, please see the attached PDF for small corrections / suggestions.
2.4 Experimental results and analysis
2.4.1 Physical phase analysis: no major issues, please see the attached PDF for small corrections / suggestions / comments and questions and clarify when necessary.
2.4.2 Micro-structure analysis: This section is clear, for the most part. Please see the comments and questions on the PDF file, especially on the references issue. The main concern that arose was the fact that the authors relate LASER power to the observation of unmelted powder, but the value of LASER energy density is not supplied or compared with literature results. This seems very important in understanding the process. I strongly suggest including this comparison.
2.4.3 Tensile performance analysis: This section is clear, for the most part. Please see the comments and questions on the PDF file, especially (again) on the references issue. As the main suggestion, I would stress the suggestion on the PDF: the authors should relate fracture morphology observations with the micro-structure observation of Fig. 6, where one can see the SLM structure disappear after the Solubilization + Ageing HT - hence, no cracks remain to hinder mechanical response?
2.4.4 Vickers hardness analysis: This section is clear, for the most part. Please see the comments and questions on the PDF file, especially (again) on the references issue. A question arises: the usage of Vickers Hardness is meant to verify the relation between mechanical properties and the observed micro-structure. A strong relation is verified. What was the size of the indentation in the micro-hardness measurement? Was it possible to measure different zones of the samples, especially the as-deposited, were one can distinguish between the melt pool and the boundaries between layers? It would be important to clarify this, as the material is quite in-homogeneous, especially in the as-deposited and annealed states.
2.5 Conclusions
2.5.1 General comments: the chapter summarizes the work and main achievements. The text is more or less clear. Since there is no “discussion” section, the reviewer feels like there should be a more detailed analysis of some of the results, not just a description.
3 Final remarks
The present work is relevant. There are important questions that arose when reading the document, and that I would like to see clarified. I feel there should be more discussion of the results and comparison with existing literature results. Sometimes the references are not used in their full potential, as they are merely mentioned. This will help increase the quality of the paper, in my view.
As suggestions for improvement, and besides addressing the comments on the PDF already mentioned, the authors should improve grammar and the formatting of the text in some places. The references should be thoroughly verified as well, as they are incorrectly attributed in many cases.

Author Response
Please find the response in the attachment

Reviewer 2 Report
This paper has studied the effects of heat treatment on the microstructure and mechanical properties of SLM 6061 aluminum alloys by comparing the three SLM 6061 samples – as deposited, annealed, and solid solution followed by aging. The results include chemical composition, surface morphology obtained with SEM, mechanical properties with uniaxial tensile testing and hardness testing, phase composition analyzed by EDS, and elemental distribution analyzed using SEM. The results have shown the microstructure changes in the heat treated samples and the resulted changes in mechanical properties. The research method is well-established. The presentations of the experimental methods are complete and clear. The paper is recommended to be published with minor changes in grammar and formatting.
Grammar mistakes in line 90 and line 234
Be consistent with aluminium and aluminum
Figure 3 caption and the following paragraph
Author Response
Thanks to your hard work and the valuable opinions. According to the modification suggestions you sent to me, the author made corresponding modifications to the paper and replied the experts' questions one by one. The specific contents are as follows.
Question 1:Grammar mistakes in line 90 and line 234.
Reply: Thanks for your suggestion.
The syntax error of Line 90 has been modified, and the modified sentence is “Among the SLM forming aluminum alloy series, SLM6061 aluminum alloy is used in the part of thermal shield device of ITER project due to its high strength at low temperature, good processing performance, and strong corrosion resistance.”
The syntax error of Line 234 has been modified, and the modified sentence is “The reason for eutectic silicon skeleton formation is related to the cooperative growth mode of the solidification process of Al-Si eutectic.”
Question 2:Be consistent with aluminium and aluminum.
Reply: Thanks for your suggestion. All aluminium and aluminum in the paper has been uniformly modified as aluminum.
Question 3:Figure 3 caption and the following paragraph.
Reply: Thanks for your suggestion. The author has changed the name of Figure 3 to “schematic diagram of heat treatment process”. The content of Figure 3 and the following paragraphs are also modified. The specific modifications are shown in the attached manuscript.
Note: The red font in the text shows the specific changes made in response to the revisions.

Reviewer 3 Report
The additive manufacturing of metallic parts is one of the fastest developing technology. The investigation of the microstructure and mechanical properties of the additively manufactured samples made from aluminum alloys is an actual topic. In the paper "Effects of Heat Treatment on Microstructure and Mechanical Properties of Selective Laser Melting 6061 Aluminum Alloy" the comparison of the different types of heat treatment on the microstructure and mechanical properties of the SLM 6061 aluminum alloys by taking thermal shield device of ITER project as engineering background was investigated. Using different microstructural technics, the authors have investigated the changes in the microstructure and mechanical properties during annealing and solution treatment. The research content of the thesis is relatively meaningful and the organizational structure of this paper is good, however, some issues should be revised.
1. Although the author indicates that "the great influence of heat treatment on the microstructure and mechanical properties of SLM aluminum alloy has been paid attention to by scholars at home and abroad", but only 4 literatures are cited, which is not convincing.
2. As for the relevant text content in Figure 6, the corresponding position in the figure is not annotated, and the author is expected to mark it accurately and clearly.
3.The silicon particles in Figure 6 (c) are not clearly visible, so it is recommended to modify them.
4. How to prove the existence of AlFeSi phase after annealing or solution aging by XRD?
5. Although EDS analysis in Figure 7-9 is point energy spectrum analysis, there is an error range. How to ensure the accuracy of data.
6. Quantitative information of microstructure is not provided in the paper, such as precipitate size in Figure 9 and dimple size in Figure 12.
7. The conclusion should be more concise.
8. It is suggested to indicate the affiliated unit of the experimental equipment.
9.Language needs to be carefully revised.
Reviewer 4 Report
The work presented by the authors is aimed at practical application. However, the authors failed to obtain high-quality samples. Some comments are in attach file.

Round 2
Reviewer 1 Report
Effects of Heat Treatment on Microstructure and Mechanical Properties of Selective Laser Melting 6061 aluminium alloys
Reviewer response to the authors reply – June 2022
Point 1: Thank your for your thorough clarifications. I would suggest including a small part of this background information on the paper, but I understand if you feel like it’s excessive.
Point 2: The clarifications provided are appropriate and eliminated the previous inconsistencies. Good work.
Points 3-5: Corrections implemented, no further issues.
Point 6: The addition of the LED parameter is a positive addition, as well as the comparison with literature. Good work.
Point 7: Suggestions implemented. Good work.
Point 8: Very good additional data on hardness. I would strongly suggest adding at least a summary of this information to section 3.2 of the paper, with the purpose of detailing the data, as currently the doubt remains about whether hardness values are affected by the zone where the measurement was performed (which it is, as demonstrated). Also, the values presented in the paper are average values, among all melt pool / melt pool boundaries?
Points 9-11: Corrections mostly implemented. Please check: reference 17 for a small glitch (Acta Materialia); and reference 26, as it is lacking the publication title.
Reviewer 4 Report
All comments were taken into account.